# Effect of Ozone and Electron Beam Irradiation on Degradation of Zearalenone and Ochratoxin A

**DOI:** 10.3390/toxins12020138

**Published:** 2020-02-24

**Authors:** Kai Yang, Ke Li, Lihong Pan, Xiaohu Luo, Jiali Xing, Jing Wang, Li Wang, Ren Wang, Yuheng Zhai, Zhengxing Chen

**Affiliations:** 1National Engineering Laboratory for Cereal Fermentation Technology, Jiangnan University, Wuxi 214122, China; yangkai164@outlook.com (K.Y.); 6160112134@vip.jiangnan.edu.cn (K.L.); yelanplh@outlook.com (L.P.); legend0318@hotmail.com (L.W.); nedved_wr@jiangnan.edu.cn (R.W.); 6190112167@stu.jiangnan.edu.cn (Y.Z.); 2Beijing Advanced Innovation Center for Food Nutrition and Human Health, Beijing Technology and Business University (BTBU), Beijing 100048, China; jwang010@126.com; 3College of Food and Pharmaceutical Science, Ningbo University, Ningbo 315000, China; 4Research Institute of Gang Yagou Healthy Food and Biotechnology, Ningbo 315205, China; 5Ningbo Institute for Food Control, Ningbo 315048, China; hellojiali77@gmail.com

**Keywords:** ozone, electron beam irradiation, degradation, zearalenone, ochratoxin A

## Abstract

Zearalenone (ZEN) and ochratoxin A (OTA) are key concerns of the food industry because of their toxicity and pollution scope. This study investigated the effects of ozone and electron beam irradiation (EBI) on the degradation of ZEN and OTA. Results demonstrated that 2 mL of 50 μg/mL ZEN was completely degraded after 10 s of treatment by 2.0 mg/L ozone. The degradation rate of 1 μg/mL ZEN by 16 kGy EBI was 92.76%. Methanol was superior to acetonitrile in terms of degrading ZEN when the irradiation dose was higher than 6 kGy. The degradation rate of 2 mL of 5 μg/mL OTA by 50 mg/L ozone at 180 s was 34%, and that of 1 μg/mL OTA by 16 kGy EBI exceeded 90%. Moreover, OTA degraded more rapidly in acetonitrile. Ozone performed better in the degradation of ZEN, whereas EBI was better for OTA. The conclusions provide theoretical and practical bases for the degradation of different fungal toxins.

## 1. Introduction

Crop pollution by fungal toxins is a global issue that causes massive annual economic losses. Large territorial areas of China are located in temperate and subtropical zones, offering favorable climatic conditions for the growth and reproduction of toxic fungi [1]. Zearalenone (ZEN) of *Fusarium* [2] and ochratoxin A (OTA) of *Aspergillus* and *Penicillium* [3] exhibited the most extensive distribution and strongest toxicity among the more than 400 toxic fungi discovered thus far [4]. Numerous studies have reported on the reproduction, genetic, and immune toxicities of ZEN, which causes nausea, vomiting, and diarrhea in humans and animals [5]. Animal toxicological experiments and clinical studies have demonstrated the strong nephrotoxicity, immunotoxicity, and carcinogenicity of OTA [6,7].

Ozone, as a strong oxidant, can attack the double bond in organic compounds through molecules and free radicals in a liquid system [8,9,10]. In addition, ozone exhibits acceptable permeability and can automatically decompose into oxygen without generating toxic residues [11]. Hence, ozonation is a fungal toxin degradation technology with promising potential, and has been the focus of numerous studies in recent decades. Xu et al. discussed the reduction of ZEN in corn flour by ozone treatment, and four ozonation products were identified by a method involving the use of ultra-performance liquid chromatography-tandem mass spectrometry [12]. Inan et al. degraded aflatoxin B_1_ (AFB_1_) in sliced and ground pimiento by 33 and 66 mg/L ozone and obtained degradation rates of 80% and 93%, respectively [13]. Zorlugenc et al. treated AFB_1_ in dry figs by ozone gas and ozonated water for 180 min, and the degradation rates of AFB_1_ were 95.21% and 88.62%, respectively [14]. These studies proved the strong degradation capability of ozonated gas and ozone water for fungal toxins.

Electron beam irradiation (EBI) is the process of irradiating products by using the electron beam generated by an electron accelerator. This technology is characterized by high energy utilization, simple operation, and safe use. EBI is a safe and effective green-processing technology. Researchers have explored the application of EBI in agricultural products and food storage, crop breeding, radiation sterilization, radiation pest control, and other emerging fields such as radiation chemical and radiation material engineering [15,16]. The degradation of fungal toxins in foods based on EBI has recently been discussed. Stepanik et al. investigated the degradation of deoxynivalenol (DON) in three production intermediates in distillers dried grain and solubles by using a dose of EBI treatment [17]. Wang et al. discussed the degradation of AFB_1_ by EBI and found that 8.6 kGy of EBI was adequate to completely degrade 5 ng/g AFB_1_ [18]. The mechanism of action was similar to that of γ-ray [19]. Water molecules were activated and ionized after EBI, generating hydroxyl radicals and hydration molecules. Hydroxyl radicals further destroy the molecular structure of the toxin.

Few studies on the degradation of ZEN and OTA by ozone and EBI are available presently, although numerous studies have focused on the degradation of AFB_1_ [20]. The effects of ozone and EBI on the degradation of ZEN and OTA cannot be easily understood due to the different structures of ZEN and OTA from AFB_1_, thereby limiting the applications of ozone and EBI in ZEN- and OTA-contaminated foods. The influences of ozone concentration, sample concentration, treatment time, and radiation dose in acetonitrile and methanol systems on the degradation rates of ZEN and OTA were discussed on the basis of previous studies on DON [21] and AFB_1_ [22] degradation by ozone and EBI. The present study provides theoretical and practical bases for OTA and ZEN degradation in food by ozone and EBI in the future.

## 2. Results

### 2.1. Standard Curves of Zearalenone (ZEN) and Ochratoxin A (OTA)

The linear correlation between the peak area and concentration (0.5–5.0 and 0.1–1.0 μg/mL) was clarified on the basis of the standard curves of ZEN and OTA (Figure 1). The regression equations of the ZEN and OTA standard curves are marked in Figure 1. R^2^ represents the square of the correlation coefficient, and the determination coefficient R^2^ is a relative index of goodness of fit between the regression line and the observed value of the sample, reflecting the proportion of the fluctuations of the dependent variable that can be explained by the independent variable. The closer R^2^ is to 1, the better the goodness of fit. R^2^ of the regression equations in the ZEN and OTA standard curves were 0.9999 and 0.9998, respectively, indicating a high degree of linear regression.

### 2.2. Degradation Rates of ZEN and OTA by Ozone

The degradation rates of 2 mL of 50 μg/mL ZEN standard working solution (a) and 2 mL of 5 μg/mL OTA standard working solution (b) by ozone under different treatment time periods are shown in Figure 2. The degradation rate of ZEN at 1 s was higher than 50%. The degradation of ZEN slowed down with the increase in treatment time. No ZEN was detected in the solution at 10 s. The degradation curve of OTA from 0–180 s was a reverse S-shaped curve. The degradation rate of OTA increased during the first 30 s and decreased between 30–60 s, but it increased gradually between 60–180 s and reached the peak (34%) at 180 s.

### 2.3. Degradation of ZEN and OTA by Electron Beam Irradiation (EBI)

#### 2.3.1. Degradation of ZEN by EBI

The degradation curves of 1.0 μg/mL ZEN in methanol and acetonitrile solution under different EBI doses (i.e., 0, 2, 4, 6, 8, 10, 12, 14, and 16 kGy) are shown in Figure 3. The degradation rate of ZEN in the acetonitrile solution was higher than that in the methanol solution under 0–6 kGy. However, the degradation rate of ZEN in acetonitrile decreased gradually with the increase in dose, higher than that in acetonitrile at an irradiation dose exceeding 6 kGy. The degradation of ZEN slowed down in methanol with the increase in irradiation dose. The degradation rates of ZEN in methanol and acetonitrile were 92.76% and 72.29%, respectively, at 16 kGy. Methanol was conducive to the degradation of ZEN by EBI at high irradiation doses. According to the literature, EBI possesses unique advantages in degrading fungal toxins, and considerable development of this approach has been complicated. Liu et al. processed AFB_1_ in sewage by EBI [23]. The 1 and 5 μg/mL toxin samples were degraded completely at 8 kGy. Peng et al. irradiated OTA in different solvent systems by electron beams and disclosed the degradation rates of OTA under the same concentration as follows: water > acetonitrile > methanol–water (60:40, v/v) [24]. Therefore, EBI can also degrade ZEN in a methanol solution.

#### 2.3.2. Degradation of OTA by EBI

The degradation of 1 μg/mL OTA in methanol and acetonitrile solution at different EBI doses (i.e., 0, 2, 4, 6, 8, 10, 12, 14, and 16 kGy) is shown in Figure 4 with S-shaped curves, indicating that the contents of the active substances (such as free radicals and active oxygen), which can react with OTA in the solvent system, initially increased and then decreased. In the irradiation dose range of 0–6 kGy, the degradation rate of OTA in the two solvent systems gradually increased and slowly degraded at 6 kGy. The degradation rates of OTA in methanol and acetonitrile at 16 kGy reached the maximum values of 84.16% and 91.56%, respectively.

#### 2.3.3. Effects of Solvents on Degradation of ZEN and OTA by EBI

Methanol is a common protic solvent and ·OH and $e_{aq}^{-}$ quencher [25]. Acetonitrile is a common aprotic polar solvent. Both are widely used as solvents of fungal toxins. The acetonitrile solutions of ZEN and OTA after EBI application turned yellow, with its color deepening as the radiation dose increased. In contrast, the methanol solution remained transparent. The acetonitrile solutions of ZEN and OTA after EBI (i.e., 0, 4.0, 8.0, 12.0, and 16.0 kGy) are shown in the order from left to right in Figure 5a,b, respectively. Kameneva et al. irradiated acetonitrile molecules in a solid inert gas matrix by x-ray [26]. The Fourier infrared spectrum detection showed that acetonitrile molecules generated CH_3_NC, CH_2_CNH, and CH_2_NCH molecular polymers and free radicals such as CH_2_CN and CH_2_NC.

## 3. Discussion

High degradation velocities of OTA in the early reaction stage may be related to the active materials in the reaction system in the reaction of degrading OTA by using ozone. Active substances were consumed completely as the reaction continued, thus resulting in the decelerated degradation. Ozone molecules degraded OTA and generated free radicals at the end of the reaction, resulting in the gradual increase in degradation speeds. This finding confirmed the research conclusions of Qi et al. [20].

The degradation rate of ZEN/Ace samples in the reaction of degrading ZEN/Met and ZEN/Ace by EBI was observably higher than that of ZEN/Met samples within 0–6 kGy. Schmelling et al. reported that organic solvents (i.e., acetonitrile and methanol solutions) treated with EBI produced less free radicals, and free radicals were in the dynamic process of generation and annihilation [27]. Guo et al. proved that methanol is a free radical scavenger [25]. In 0–6 kGy, free radicals generated by EBI in acetonitrile can be used for ZEN degradation, whereas free radicals in methanol were eliminated by methanol molecules once generated, so the degradation rate of ZEN in methanol was low. At 6–16 kGy, the degradation rate of both ZEN samples slowed down, and the degradation rate of the ZEN/Met samples was higher than that of the ZEN/Ace samples. A possible reason for this trend was that at 6–16 kGy, the degradation of ZEN in acetonitrile resulted in a sharp decline in its concentration. Acetonitrile generated a large amount of free radicals by irradiation that slowed down the degradation rate of ZEN/Ace. In the methanol system, the free radical-scavenging effect of methanol was limited and relatively more free radicals were present in the solution. Thus, the degradation rate of ZEN/Met was still higher than that of ZEN/Ace.

Within the irradiation dose range of 0–16 kGy, the degradation rate of OTA in the acetonitrile solution exceeded that in the methanol solution, contradicting the results of Peng et al. [24]. This result may be due to different processing conditions such as volume, container, and handling operations. This study demonstrated that EBI can effectively degrade OTA in methanol and acetonitrile with a higher degradation rate in acetonitrile. On one hand, the free radicals of degrading OTA were reduced because of the free radical-scavenging effect of methanol. On the other hand, the molecular structure of OTA contained more free radical-charged sites, and acetonitrile free radicals produced by EBI were still relatively more in the degrading process, thus the phenomenon of a ZEN/Ace degradation rate lower than that in ZEN/Met at 6–16 kGy was not observed.

Moreover, different substrates can considerably affect the reaction. High-performance liquid chromatography (HPLC) chromatograms of ZEN and OTA degradation by a 3 kGy EBI dose in acetonitrile solution were compared with those in methanol (Figure 6). Within the retention time range of 1.5–5 min, the HPLC chromatograms of ZEN and OTA in acetonitrile exhibited numerous absorption peaks, whereas no absorption peaks were observed in the methanol solution and the blank group. The degradation curves of ZEN and OTA were remarkably different under the same conditions, indicating the considerable difference of their reaction mechanisms. This difference can be used to interpret the reaction results of ZEN and OTA in different solvents. In addition, this difference may be partially attributed to the different molecular structures of ZEN and OTA.

## 4. Conclusions

Ozone and EBI were proven to be a safe, efficient, and environmentally friendly processing method in degrading ZEN and OTA. In this study, the degradation rate of the ZEN and OTA standard working solutions increased with the extension of ozone treatment time. ZEN was completely degraded within 10 s at 2 mg/L of ozone concentration, and the degradation rate of OTA was 34% at 180 s by ozone treatment, suggesting that ZEN was more sensitive to ozone. The degradation rate of ZEN and OTA solutions by EBI increased with the irradiation dose. At high radiation dose, methanol was beneficial to the degradation of ZEN by EBI and acetonitrile was more beneficial to the degradation of OTA by EBI. EBI can induce the reaction among acetonitrile molecules and between acetonitrile and toxin molecules by comparing the degradation rates of ZEN and OTA in different solutions using HPLC chromatograms. However, the high cost of equipment limits the wide application of ozone treatment and EBI. In addition, a high EBI dose on grain will affect the color of the product, which is not conducive to the subsequent processing and utilization. Further research needs to be done to investigate the mechanism of degradation of mycotoxin by ozone and EBI and discuss the different process conditions and instrument parameters in degrading mycotoxins in cereals and cereal-derived food products.

## 5. Materials and Methods

### 5.1. Materials and Instruments

#### 5.1.1. Materials and Reagent

Standard ZEN and OTA products (purity ≥ 99.8%) and acetic acid (HPLC grade) were obtained from J&K Scientific Ltd. (Shanghai, China). HPLC-grade methanol (Met) and acetonitrile (Ace) were purchased from Fisher Scientific Company (Waltham, MA, USA). Milli-Q quality water (resistance ≥ 18.2 MΩ/cm) was prepared by a Millipore-QSP ultrapure water instrument (Millipore, Bedford, MA, USA). Nitrogen (purity ≥ 99.8%) and oxygen (purity ≥ 99.8%) were obtained from Wuxi Xinnan Chemical Gas Co., Ltd., Wuxi, China. The other analytical pure reagents were purchased from China Pharmaceutical Chemical Reagent Co., Ltd., Shanghai, China.

#### 5.1.2. Major Apparatus

HPLC (HPLC 1260) with a fluorescence detector and ZORBAX SB-C18 column was purchased from Agilent Technologies (Palo Alto, CA, USA). The AB5.0 electron beam accelerator was obtained from Wuxi EL PONT Radiation Technology Co., Ltd. (Wuxi, China) and the CFG-3-20 g-type ozone generator was purchased from Qingdao Guolin Industrial Co., Ltd. (Qingdao, China). An Ideal 2000 Ozone concentration analysis recorder was obtained from Zibo ADEL Measurement Control Co., Ltd. (Zibo, China) and the MD200-1 pressure-blowing instrument was purchased from Hangzhou Allsheng Instrument Co., Ltd. (Hangzhou, China).

### 5.2. Experimental Methods

#### 5.2.1. Preparation of Standard Stock and Working Solutions of ZEN and OTA

ZEN was dissolved in a methanol solution, and OTA was dissolved in an acetonitrile solution to prepare 100 and 10 μg/mL standard stock solutions, respectively. The solutions were stored at −20 °C. A certain amount of ZEN standard stock solution was collected and dried using nitrogen. Then, methanol and acetonitrile were added for secondary dissolution to prepare 1.0 μg/mL ZEN/Met and ZEN/Ace standard working solutions, respectively. The solutions were stored at 4 °C for EBI. A certain amount of ZEN standard stock solution and methanol was used to prepare 50 μg/mL of standard working solution and stored at 4 °C for subsequent ozone treatment. A certain amount of OTA standard stock solution was obtained and dried by nitrogen. Then, methanol and acetonitrile were added for secondary dissolution to prepare 1.0 μg/mL OTA/Met and OTA/Ace standard working solutions, respectively. The solutions were stored at 4 °C for EBI. A portion of OTA standard stock solution and acetonitrile was used to prepare a 5 μg/mL working solution and stored at 4 °C for subsequent ozone treatment.

#### 5.2.2. Measurement of ZEN and OTA Contents

An Agilent 1260 HPLC with a G1321B fluorescence detector was employed. The chromatographic column was a ZORBAX SB-C18 column (4.6 mm × 150 mm). The filling diameter and column temperature were 5 μm and 35 °C, respectively. The injection volume was 20 μL. The flow phase of ZEN was methanol/water (60/40, v/v) with the flow rate set at 1.0 mL/min. The detection wavelength was 274 and 440 nm for the excitation and emission wavelengths, respectively. The flow phase of OTA was water/acetonitrile/acetic acid (56/43/1, v/v/v), and the flow rate was 0.9 mL/min. The detection wavelength consisted of 333 and 477 nm for the excitation and emission wavelengths, respectively.

#### 5.2.3. Drawing of Standard ZEN and OTA Curves

Several ZEN and OTA standard stock solutions were selected. ZEN (i.e., 0.5, 1.0, 2.0, and 5.0 μg/mL) and OTA standard working solutions (i.e., 0.1, 0.2, 0.5, and 1.0 μg/mL) were prepared by the flow phase. The relation curves between the absorption peak area in the liquid chromatograph and concentration of solutions were drawn and the standard curves of ZEN and OTA were drawn according to these relation curves.

#### 5.2.4. Degradation of ZEN and OTA by Ozone

Ozone was generated by a high-pressure discharger from an ozone generator connected with an external oxygen source. Ozone concentration was adjusted by controlling the current of the ozone generator, and changes in ozone concentration were monitored online by ozone concentration detectors. The excess ozone was eliminated by the decomposition of the ozone destroyer.

A total of 2 mL ZEN (50 μg/mL) and OTA (5 μg/mL) working solutions were dissolved in a piece of 10 mL polyethylene centrifuge tube, which was supplied with ozone. The ozone treatment conditions for ZEN were as follows: concentration = 2.0 mg/L, flow rate = 1.0 L/min, and treatment time = 0, 1, 2, 3, 5, and 10 s. The ozone treatment conditions for OTA were as follows: concentration = 50.0 mg/L, flow rate = 1.0 L/min, and treatment time = 0, 10, 30, 60, 90, 120, and 180 s. After the ozone treatment, nitrogen was supplied for 3 min, and then the reaction was terminated. Subsequently, 1 mL flow phase was used for secondary dissolution.

#### 5.2.5. Degradation of ZEN and OTA by EBIs

A total of 2 mL ZEN and 2 mL OTA working solutions were placed in a 5 mL polyethylene centrifuge tube. Irradiation doses were 0, 2, 4, 6, 8, 10, 12, 14, and 16 kGy. The accelerated electron energy was 5 MeV, and the electron beam current was 20 mA with a 1000 mm scan width. The dose rate was 2 kGy/s. The samples were dried by nitrogen after irradiation, and a 1 mL flow phase was used for secondary dissolution.

#### 5.2.6. Data Processing

Sample processing and detection were repeated at least three times. One-way analysis of variance was performed by the Statistical Package for the Social Sciences version 17.0. *p* < 0.05 was considered to be statistically significant. *p* >0.05 was not statistically significant.

## Figures and Tables

**Figure 1 toxins-12-00138-f001:**
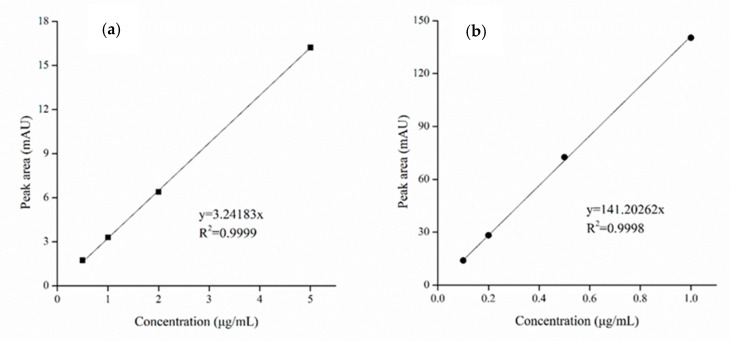
High-performance liquid chromatography (HPLC) standard curves of zearalenone (ZEN) (**a**) and ochratoxin A (OTA) (**b**).

**Figure 2 toxins-12-00138-f002:**
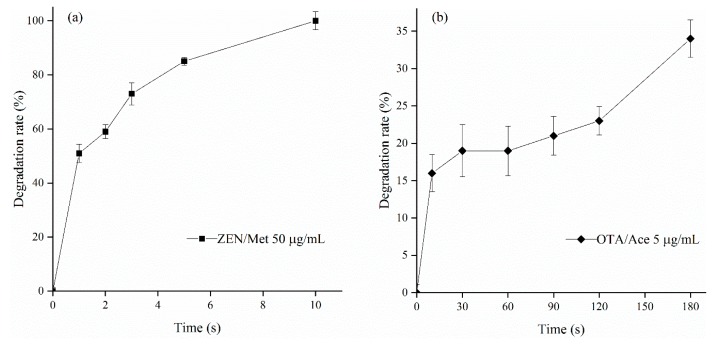
Degradation curves of ZEN (**a**) and OTA (**b**) by ozone at different treatment time periods.

**Figure 3 toxins-12-00138-f003:**
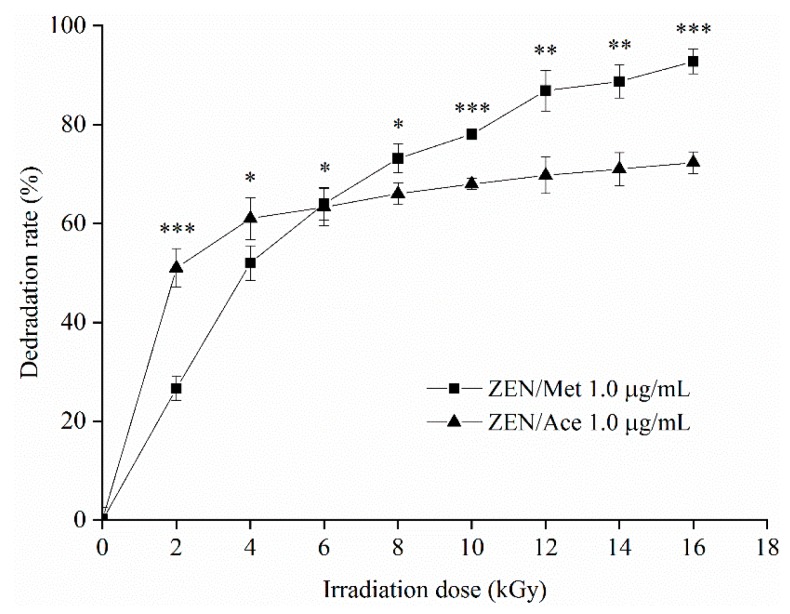
Degradation curve of ZEN in methanol (Met) and acetonitrile (Ace) at different electron beam irradiation (EBI) doses. Data are presented as means ± standard deviation (SD). *** *p* < 0.01, ** 0.01 < *p* < 0.05, and * *p* > 0.05.

**Figure 4 toxins-12-00138-f004:**
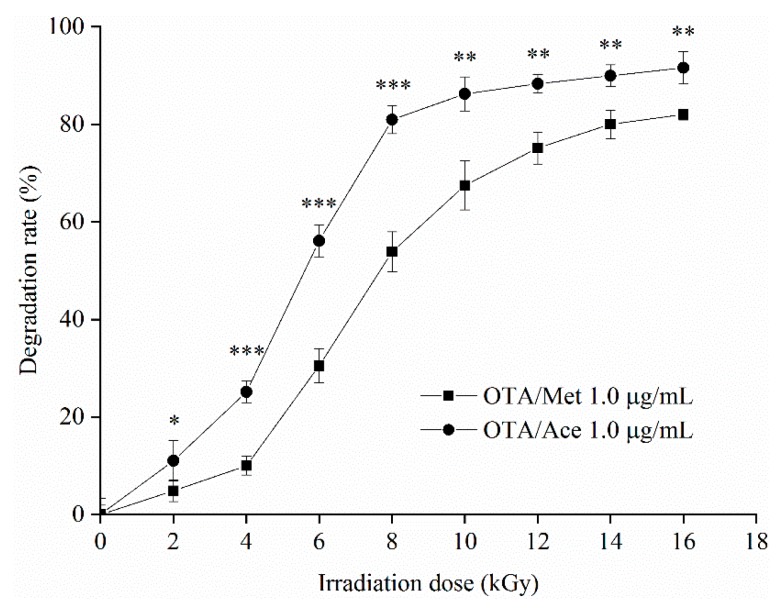
Degradation curve of OTA in Met and Ace at different EBI doses. Data are presented as means ± SD. *** *p* < 0.01, ** 0.01 < *p* < 0.05, and * *p* > 0.05.

**Figure 5 toxins-12-00138-f005:**
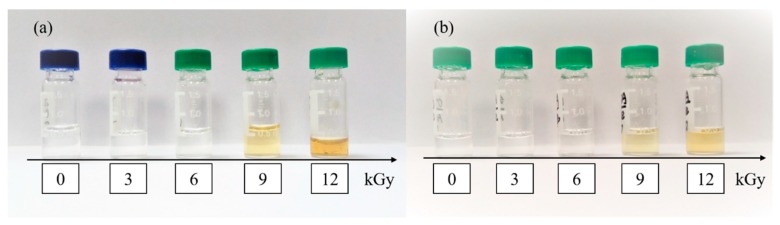
Degradation samples of ZEN (**a**) and OTA (**b**) treated with EBI in Ace.

**Figure 6 toxins-12-00138-f006:**
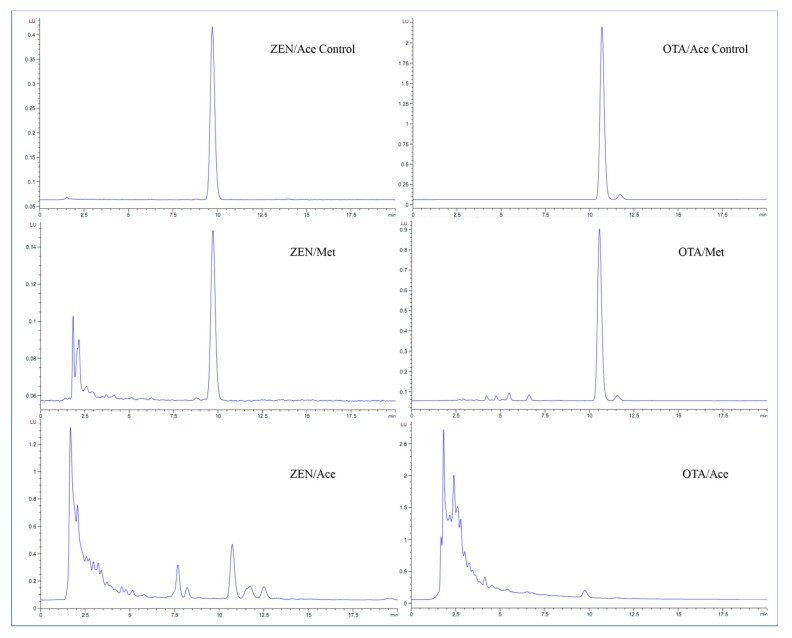
HPLC chromatograms of ZEN and OTA treated with 3 kGy EBI dose in Met and Ace.

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
