# Peer review of "Effect of Ozone and Electron Beam Irradiation on Degradation of Zearalenone and Ochratoxin A"

_toxins, 2020, doi:10.3390/toxins12020138_

Round 1

Reviewer 1 Report

In this manuscript the authors provide an interesting study investigating the effect of ozone or electron beam irradiation (EBI) on degradation of mycotoxins, namely zearalenone (ZEA) and ochratoxin A (OTA). Although the efficiency of these techniques have already been investigated to reduce the amount of several mycotoxins in feeds and foodstuffs, only few studies very done with ZEA and OTA. This manuscript deals with the degradation of these mycotoxins in a pure chemical system.

The Abstract is well-written and provides an accessible summary of the paper.

The Introduction section is a well-written part of this manuscript providing important literature on the background of this topic. Although – as the authors also mentioned - few studies are available dealing with this topic, I miss from the references the article written by Stepanik et al. (2007), dealing with the degradation of DON by EBI, and the one by Xu et al. (2019) investigating the degradation of ZEA by ozone.

I advise to use the term 'ozonated water' instead of 'ozone water' (L39, L41).

The methodology of the experiment is correct, but the weakest part of the manuscript is the statistics.

- The authors mention at L210 that they performed ANOVA for data analysis, but no significant differences are shown on diagrams (Fig. 3-4)!

- Were there any significant differences at the end of the EBI treatment between the two different solutions of mycotoxins?

- If there are two different treatments why was ANOVA used instead of t-test?

- What others are shown on the diagrams (Fig. 3-4) next to the mean values (SD or SEM)?

- There is no information about Data processing sub-chapter about the calculation of the standard curves of ZEA and DON.

- There is no need to mention the equations of these standard curves twice (in text and on diagram).

- I advise to show the investigated concentrations of the mycotoxins in the figure legends.

- At Figure 3,4 use even scale (0,2,4 …) on the x-axis.

- How were the applied ozone concentrations chosen in case of ozone treatment? Why was it 25-times higher in case of OTA compared to ZEN?

Some minor changes need to be corrected.

L46: correct: and as insecticide L70: correct: periods are shown L73 – This sentence is not true in this form. Correct: The degradation rate of OTA increased during the first 30 s and decreased between 30-60 s, … L88: correct: is shown L95: correct: electron L96: correct: are widely used as solvents of fungal toxins L217: use: ochratoxin-A L220: add more information: 497- L227: correct the name of authors – now it is meaningless L259: use: Part A

The Discussion of the obtained results is correct, but I found the Conclusion part too general (mainly the first two sentences).

I also miss from the manuscript mentioning the possible disadvantages of these treatments (e.g. cost of these methods vs. efficacy, potential nutritional or organoleptic changes in the feedstuffs or foods).

Overall, the study is interesting and provides new data about degeneration of OTA and ZEA (dissolved in different solvents) by ozone or EBI treatments in pure chemical system.

Reviewer 2 Report

The paper presents the results of two different degradation methods (ozone and electron beam irradiation) applied on two toxic fungi, ZEN and OTA, obtained in standard form from a producer - J&K Scientific Ltd. (China). The work is relatively well conceived, but the novelty is not well emphasized and argued by comparing with data from the literature.

            Some remarks are to be done:

English language must be improved! Rows 42 to 46: The EB irradiation technology description must be improved. Please, rewrite the end of the paragraph. EB irradiation cannot be used as insecticide! Indeed EB irradiation can prevents or stops the spread of insects, depending on many factors. There are many works that can help the authors to do that. For example:

- E. Manaila, M.D. Stelescu, G. Craciun, Daniel Ighigeanu, Wood Sawdust/Natural Rubber Ecocomposites Cross-Linked by Electron Beam Irradiation, Materials 2016, 9, 503 (see in Introduction).

- M.D. Stelescu, E. Manaila, G. Craciun, D. Ighigeanu, Electron Beam Processing of Ethylene- PropyleneTerpolymer-Based Rubber Mixtures, International Journal of Chemical and Materials Engineering 2018, 12(6), 263-267 (see in Introduction).

Rows 64 to 65: It is not necessary to specify what was plotted on X and Y axis! It can be seen from Fig 1. Rows 65 to 67: What is the relevance of the regression equations values? I think they need to be explained. Rows 68 to 75: The chemical formulas of the molecules and free radicals listed here must be written using the subscript facility (CH3NC not CH3NC and so on). Rows 122 to 142: The authors should explain the obtained results and offer their own interpretation of the obtained results, by correlating them with the conditions under which they were obtained and with specialty literature. The entire chapter must be improved. Rows 143 to 151: The conclusions of the paper must be improved. The results obtained and the connection between them and the statements made are not mentioned at all: “The results presented in this 148 study demonstrated ozone and EBI are promising in degradation of ZEN and OTA, which confirmed 149 its potential in decontamination of mycotoxins.” Rows 196 to 203: The subchapter 5.2.4. Degradation of ZEN and OTA by ozone must be improved. The authors did not mention anything about the installation in which the ozone degradation experiments were done (a laboratory installation, a standard one?). Rows 204 to 208: The subchapter 5.2.5. Degradation of ZEN and OTA by EBIs must be improved. The authors did not mention anything about radiation dosimetry, an essential aspect in radiation processing. The EB accelerator parameters are insufficiently described. It is not clear who the parameter 20 mA is (beam current?). There are works that can help the authors to do that. For example:

- G. Craciun, E. Manaila, D. Ighigeanu, New Type of Sodium Alginate-g-acrylamide Polyelectrolyte Obtained by Electron Beam Irradiation: Characterization and Study of Flocculation Efficacy and Heavy Metal Removal Capacity, Polymers 2019, 11, 234;

(see in the subchapter 2.2 Experimental Installation and Sample Preparation)

      The EB irradiation was made in the ampoules that can be seen in Fig 5? In my opinion they are not suitable.

Rows 210 to 211: Please define SPSS 17.0. and P

The paper can be recommended for publication, but after a serious revision.

Round 2

Reviewer 2 Report

 I recommend the publication of the paper in present form.